# Body Image Satisfaction, Food Consumption, Diet Quality, and Emotional Management in Adolescence: A Longitudinal Analysis from the SI! Program for Secondary Schools Trial

**DOI:** 10.3390/nu17243882

**Published:** 2025-12-12

**Authors:** Patricia Bodega, Juan M. Fernández-Alvira, Amaya de Cos-Gandoy, Luis A. Moreno, Mercedes de Miguel, Carla Rodríguez, Jesús Martínez-Gómez, Emily P. Laveriano-Santos, Sara Castro-Barquero, Ramón Estruch, Rosa M. Lamuela-Raventós, Rodrigo Fernández-Jiménez, Gloria Santos-Beneit

**Affiliations:** 1Foundation for Science, Health, and Education (SHE), 08008 Barcelona, Spain; 2Centro Nacional de Investigaciones Cardiovasculares (CNIC), 28029 Madrid, Spain; 3GENUD (Growth, Exercise, NUtrition, and Development) Research Group, Faculty of Health Sciences, Instituto Investigación Sanitaria de Aragón (IIS Aragón), Instituto Agroalimentario de Aragón (IA2), University of Zaragoza, 50009 Zaragoza, Spain; lmoreno@unizar.es; 4Consorcio CIBER, Fisiopatología de la Obesidad y Nutrición (CIBERObn), Instituto de Salud Carlos III (ISCIII), 28029 Madrid, Spain; 5Institute for Global Health (ISGlobal), 08003 Barcelona, Spain; 6Institute of Nutrition and Food Safety (INSA-UB), University of Barcelona, 08921 Santa Coloma de Gramenet, Spain; 7Department of Internal Medicine, Hospital Clínic, Institut d’Investigacions Biomèdiques August Pi I Sunyer (IDIBAPS), University of Barcelona, 08036 Barcelona, Spain; 8Polyphenol Research Group, Department of Nutrition, Food Science and Gastronomy, School of Pharmacy and Food Sciences, Barcelona University, 08028 Barcelona, Spain; 9Departament of Cardiology, Hospital Universitario Clínico San Carlos, Instituto de Investigación Sanitaria San Carlos (IdISSC), 28040 Madrid, Spain; 10Centro de Investigación Biomédica En Red en enfermedades CardioVasculares (CIBERCV), 28029 Madrid, Spain

**Keywords:** body acceptance, dietary habits, self-esteem, emotional eating, teenagers, nutritional status

## Abstract

**Background**: Adolescence involves physical and psychological changes that often conflict with body ideals, potentially increasing body image (BI) dissatisfaction and unhealthy diet. The main objective was to analyze cross-sectional and longitudinal associations between BI satisfaction (BIS) and nutritional status, dietary habits (DH), and emotional management (EM) in adolescents. **Methods**: 1315 adolescents from the SI! Program for Secondary Schools trial reported their BIS, DH, and EM at ages 12, 14 and 16. Linear and logistic mixed models estimated the associations between BIS, food intake, diet quality, and EM. Linear and logistic mixed-effect models for repeated measures assessed mean change estimates from baseline to 2- and 4-year follow-up in food intake and EM, stratified by gender. **Results**: A large proportion of adolescents without excess weight were dissatisfied with their BI. BIS was significantly associated with DH, especially in boys. Body-satisfied adolescents at baseline tended to have healthier DH (diet quality index: boys 56.7 ± 13.0; girls 58.8 ± 12.7) and showed a greater improvement in diet quality at 16 years (boys 3.44 (1.50, 5.37); girls 1.85 (0.18, 3.52)). Adolescents who desired to lose weight decreased their intake of sugar-sweetened beverages, sweets, and snacks, whereas boys who desired to gain weight consumed more frequently fast food, sweets, snacks, and processed meat. Body-satisfied adolescents had higher self-esteem, and girls desiring to gain weight presented higher emotional eating. **Conclusions**: BIS was associated with nutritional status, and EM, showing gender differences. Overall, the desire to gain weight was associated with unhealthier DH. Educational interventions should promote self-esteem and BIS by focusing messages on healthy eating instead of body weight.

## 1. Introduction

Adolescence is a transitional stage of physical and psychological maturation in which self-image is consolidated and lifestyle habits, especially dietary behaviors, become increasingly autonomous [1,2,3,4,5]. Scientific interest in body image (BI) has increased since the 1980s, parallel to growing concerns about nutritional status, lifestyle behaviors, and eating disorders [1]. BI develops within an individual’s personal history and is shaped by sociocultural models, ideals, and values attributed to the body [2,3,4,5]. Body image satisfaction (BIS), the attitudinal component of BI, reflects the discrepancy between perceived and desired body shape and size [6]. BI concerns are widespread across populations and tend to emerge early, often showing higher prevalence among girls, although evidence regarding gender differences is inconsistent [3,4,5,7,8]. Both distortion and dissatisfaction may persist into adulthood, even though BI remains a dynamic construct over time [1,7,8].

Nutritional status plays an important role in BI development. Societal pressure to achieve appearance ideals contributes to inadequate food intake, extreme weight-control behaviors, and the onset of eating disorders. The pursuit of an aesthetically ideal body often motivates restrictive dieting behaviors, whereas dissatisfaction is also associated with excess weight and physical inactivity [1,4,5,7,9]. Conversely, active lifestyles appear to mitigate BIS, partially due to body changes related to sports practice, such as increased muscularity or leanness [1,10].

BI concerns are closely interconnected with psychological well-being. More negative BI is associated with reduced self-esteem, greater emotional vulnerability, and increased emotional eating, defined as the use of food to cope with stress and characterized by greater consumption of energy-dense, palatable foods [10,11,12,13]. Self-esteem, a subjective evaluation of personal worth and efficacy, functions as a protective factor against anxiety and depression [11,14,15]. However, social pressure, beauty standards, external expectations, and excessive screen-based behaviors can adversely affect adolescents’ self-esteem, particularly among girls and those with overweight or obesity, with effects that may persist into adulthood [11,14,16]. Although emotional eating has been linked to eating disorders, depression, and low self-esteem, its association with nutritional status remains inconsistent, and evidence on its relationship with BI dissatisfaction in healthy adolescents is limited [17,18,19].

Despite the relevance of these factors, there is a notable lack of longitudinal studies examining how BIS influences dietary habits and emotional management in healthy adolescent populations, with most evidence coming from adults or individuals with overweight or eating disorders [20].

The present study aims to address this gap by (1) describing BIS across nutritional status categories; (2) examining associations between BIS and food consumption and diet quality; and (3) exploring the interplay between BIS, emotional eating, and self-esteem in adolescence.

## 2. Methods

### 2.1. Study Design and Subjects

The longitudinal data used in this article comes from the SI! (*Salud Integral*) Program for Secondary Schools trial, a prospective cohort study carried out from 2017 to 2021 in 24 public Secondary Schools in the metropolitan areas of Barcelona and Madrid, Spain (NCT03504059, 9 January 2018) (https://clinicaltrials.gov/study/NCT03504059 (accessed on 1 October 2025)). 

The study was approved by the University of Barcelona Bioethics Committee (code IRB00003099, approval date 12 April 2016), the Committee for Ethical Research of the Fundació Unió Catalana d’Hospitals in Barcelona (code CEI 16/41, approval date 25 May 2016), and the Committee for Ethical Research of the Instituto de Salud Carlos III in Madrid (code CEI PI 35_2016-v2, approval date 20 June 2016). All participants gave written informed consent. The study design, sample size calculation, randomization of the schools, and main results of the trial have been published previously [21,22]. A total of 1326 adolescents were assessed at the beginning of 1st grade of Secondary School (baseline, ~12 years old). Follow-up assessments after 2 years (~14 years old) and 4 years (~16 years old) were performed at the end of 2nd and 4th grade, respectively. Given the close relationships between body image dissatisfaction (BID), eating disorders, and emotional management [23,24], adolescents who reported eating disorders (*n* = 4) or emotional disorders such as depression or anxiety (*n* = 6) at any time were excluded from the analyses. Additionally, because of the associations among transgender individuals, body dysmorphia, BID, and eating disorders [25], adolescents being in a gender transition process (*n* = 1) were also excluded from the current analyses. The analysis thus included 1315 participants at baseline. According to the SI! Program protocol, participants were followed approximately at 2 and 4 years, with retention rates of 1234 (93.1%) and 1125 (84.8%) participants, respectively [22].

The results reported in this manuscript adhere to the STrengthening the Reporting of OBservational studies in Epidemiology-Nutritional Epidemiology (STROBE-nut) Statement checklist (Appendix A) [26].

### 2.2. Body Image Satisfaction

BIS was assessed with the Stunkard Figure Rating Scale [27,28,29]. Participants were shown body silhouettes and asked to mark the one they most identified with (perceived BI) and the one they would most like to resemble (desired BI). BIS was calculated by subtracting the perceived silhouette from the desired silhouette. Negative values indicate a desire to lose weight, positive values indicate a desire to gain weight, and zero indicates satisfaction with BI. Missing values were not imputed (baseline, *n* = 5 (0.4%); 2-year follow-up, *n* = 26 (2.1%); and 4-year follow-up, *n* = 29 (2.6%)).

### 2.3. Dietary Assessment

Participants answered a food frequency questionnaire (Children’s Eating Habits Questionnaire, CEHQ) related to their food consumption during the 4 weeks preceding each assessment. This questionnaire was previously validated in the frame of the IDEFICS study [30,31,32]. A conversion factor ranging from 0 (‘I have no idea’ or ‘missing data’) to 30 (‘≥4 times/day’) was used to transform answers into weekly consumption frequencies. According to the validation study, it was assumed that the consumption frequency as reported in the questionnaire could be equated to the number of servings [30]. Participants with more than 50% of missing answers or questions answered ‘I have no idea’ were excluded from the analysis (baseline = 1 (0.8%), 2-year follow-up = 102 (7.8%), 4-year follow-up = 207 (15.7%)).

Eight food groups were defined by grouping 24 food items [30,31,32] as follows: 1. Fruits and vegetables (cooked and raw vegetables, potatoes, beans, and fresh fruits without added sugar); 2. Fast food (fried potatoes, pizza as main dish, hamburgers, hot dogs, kebabs, wraps, and falafel); 3. Sugared and sweetened beverages (packaged fruit juice, sweetened and diet drinks, and sweetened milk); 4. Sweets (chocolate or nut-based spreads, savory pastries, fritters, chocolate, candy bars, candies, marshmallows, biscuits, packaged cakes, pastries, puddings, ice cream, and milk or fruit-based bars); 5. Snacks (crisps, maize crisps, and popcorn); 6. Nuts (nuts, seeds, and dried fruits); 7. Whole grains (porridge, oatmeal, cereals, muesli, unsweetened, whole meal bread, dark rolls, and dark crispbread); and 8. Processed meat (cold cuts, preserved and ready-to-cook meat products).

Additionally, the Diet Quality Index for Adolescents (DQI-A) (ranging from −33 to 100%) was calculated according to a previously described protocol [33,34]. The DQI-A includes three domains: *Dietary quality* (−100 to 100%), based on a ‘preferable group‘, an ‘intermediate group‘, and a ‘low-nutrient, energy-dense group‘ (Appendix A); *Dietary diversity* (0 to 100%), representing the degree of variation in the diet; and *Dietary equilibrium* (0 to 100%), representing the difference between the adequacy and the excess based on current Spanish recommendations [35,36]. The final score was calculated as the mean of the three domains. A higher DQI-A score indicates higher compliance with dietary guidelines.

### 2.4. Emotional Management

Self-esteem was assessed with a subscale of the Child Health and Illness Profile-Adolescent Edition questionnaire [37]. Emotional Eating was assessed with the corresponding section of the Three Factor Eating Questionnaire-R18 [38]. In both cases, scores were calculated only when all questions were answered, and high scores correspond to high self-esteem and to a high degree of emotional eating.

### 2.5. Nutritional Status

Nutritional status was assessed from body mass index (BMI), calculated as body weight divided by height squared (kg/m^2^). Body weight and height were measured with a body composition scale (OMRON BF511, OMRON HEALTHCARE Co., Muko, Kyoto, Japan) and a stadiometer (Seca 213, Hamburg, Germany), respectively. Age- and sex-adjusted BMI z-scores were calculated according to Centers for Disease Control guidelines [39]. Underweight, normal weight, overweight, and obesity were defined as BMI <5th percentile, <85th percentile, between 85th and 95th percentiles, and >95th percentile, respectively. Missing values were not imputed (baseline, *n* = 9 (0.7%); 2-year follow-up, *n* = 42 (3.4%); and 4-year follow-up, *n* = 46 (4.1%)).

### 2.6. Covariates

Moderate-to-vigorous physical activity was calculated following the protocol previously described [22]. Adolescents wore an accelerometer (Actigraph wGT3X-BT, ActiGraph, Pensacola, FL, USA) on their non-dominant wrist for 7 consecutive days. Records were considered valid if they provided data from a minimum of four consecutive or non-consecutive days, with at least 600 min per day of wear time [40]. Information from the validated QAPACE survey (*Quantification de L’Activité Physique en Altitude chez les Enfants*) [41] was used to quantify individual moderate-to-vigorous physical activity when accelerometer information was not available. Sexual maturation stage (from I to V) was self-reported by participants with the support of pictograms [42].

Sociodemographic information was collected through the family questionnaire. A migrant background was assumed when one or both parents were born outside Spain. Educational level was defined as the highest reported in the household and was classified according to the International Standard Classification of Education (ISCED) as low (secondary studies or below, ISCED levels 0 to 3), intermediate (postsecondary non-tertiary education or short-cycle tertiary education, ISCED levels 4 to 5), and high (university studies, ISCED levels 6 to 8) [43]. The information on migrant status and parental educational level used for analysis was that obtained at baseline, or, if this was unavailable, data collected in subsequent follow-up assessments was used. If no information was available from any of the assessments, migrant status and education level were classified as unknown.

### 2.7. Statistical Methods

Statistical differences were identified by chi-square test for categorical variables and by unpaired t-test for continuous variables. Cross-sectional differences in food intake, DQI-A, and emotional management (self-esteem or emotional eating) according to BIS were assessed with linear mixed-effect models at baseline and 2- and 4-year follow-up. Fixed effects were BIS, age, parental education level (low/intermediate/high/unknown), nutritional status at baseline (underweight/normal weight/overweight/obesity), migrant background (yes/no/unknown), moderate-to-vigorous physical activity and sexual maturity status, and randomization group (control/short intervention/long intervention). Region (Madrid/Barcelona) and schools were handled as random effects. Bonferroni correction (*p* ≤ 0.05) was used to adjust for multiple pairwise comparisons. Additionally, tertiles of food consumption, DQI-A, self-esteem, and emotional eating were created given the lack of established cut-off points, and multilevel mixed-effects logistic regression models were applied to assess the odds of being in the highest tertile of the dependent variable according to BIS in each assessment. In these models, the same fixed effects and random effects described above were applied. Longitudinal differences in food intake, DQI-A, and emotional management according to BIS were estimated using linear and logistic mixed-effect models for repeated measures at baseline and 2- and 4-year follow-up, applying the same fixed and random effects described but adding participants as an additional random effect. Lastly, associations between changes in food consumption or emotional management variables and BIS during adolescence (4-year follow-up vs. baseline) were estimated using linear mixed-effect models, applying the same fixed effects described above plus food consumption or emotional score at baseline (continuous variable) as required. Region and schools were handled as random effects.

All analyses were performed with complete cases and stratified by gender. Statistical significance was set at a threshold of *p* ≤ 0.05. All analyses were conducted with Stata version 15 (StataCorp, College Station, TX, USA).

## 3. Results

A total of 1315 adolescents (48.2% girls) aged 12.5 (±0.4) years at baseline were included in the analysis. The number of participants at 2-year follow-up was 1234 [13.9 (±0.4) years, 47.6% girls], and the number at 4-year follow-up was 1125 [15.8 (±0.4) years, 48.4% girls]. No differences in baseline sociodemographic characteristics (region, migrant background and parental education) were found according to gender. Significant differences were found in nutritional status and BIS according to gender (*p*-value < 0.001). A higher proportion of boys (32.4%) than girls (22.1%) presented with excess weight (overweight and obesity), and a higher proportion of boys were dissatisfied with their BI (75.0% boys vs. 57.7% girls). Girls showed a higher DQI-A score (girls: 58.8 (12.7) vs. boys: 56.7 (13.0), *p*-value = 0.004) and a higher quality score (girls: 45.7 (23.6) vs. boys: 38.9 (25.1), *p*-value < 0.001). No differences were found in baseline self-esteem and emotional eating (Table 1).

### 3.1. Body Image Satisfaction and Nutritional Status

The distribution of BIS throughout the study according to nutritional status (BMI classification percentile) is shown in Figure 1. Only 12.0% of adolescents were satisfied with their BI in all three assessments. At 4-year follow up, the proportion of girls desiring to lose weight increased, along with the proportion of boys who desired to gain weight (Appendix A). On average, nearly half of adolescents without excess weight were unsatisfied with their BI, while most adolescents with excess weight expressed a desire to lose weight. The percentages of boys without excess weight who desired to gain weight and girls without excess weight who desired to lose weight increased with age, especially between the 2- and 4-year follow-up assessments. The proportion of boys with excess weight who were satisfied increased over time, while for girls this parameter did not change.

### 3.2. Body Image Satisfaction and Diet

BIS was significantly associated with differences in the consumption of several food groups in both boys and girls (Table 2 and Table 3). Boys who desired to lose weight were less likely to be in the highest consumption category for sugared and sweetened beverages and sweets. Conversely, boys who desired to gain weight had higher frequencies of consumption of fast food, sweets, snacks, and processed meats than boys who desired to lose weight and were more likely to be in the highest consumption category for fast food, sweets, and snacks than boys who were satisfied with their BI. No differences in food intake in relation to BIS were found among girls.

The DQI-A improved in both genders after 4 years of follow-up (Appendix A), mainly due to the *dietary quality* component, with boys satisfied with their BI showing the most improvement. Adolescents desiring to gain weight showed no significant changes in the DQI-A, although girls desiring to gain weight showed the most pronounced worsening of the *dietary equilibrium* component. Boys satisfied with their BI decreased their intake of unhealthy foods (sugared and sweetened beverages and processed meat), whereas boys dissatisfied with their BI decreased the consumption of fruits and vegetables, sugared and sweetened beverages, whole grains, and processed meat. Boys desiring to lose weight also decreased their intake of fast food, sweets, and snacks (Figure 2a). The analysis also revealed that body-satisfied girls and girls desiring to lose weight significantly reduced their consumption of unhealthy foods (fast food, sugared and sweetened beverages, sweets, snacks, and processed meat) and that body-satisfied girls increased the consumption of fruits and vegetables compared to girls desiring to lose weight (Figure 2b). DQI-A increased in body-satisfied adolescents over time, whereas the score in girls who desired to gain weight was significantly lower than that of their counterparts (Figure 2). Cross-sectional analysis is presented in Appendix A, and the results from unadjusted models regarding associations between BIS and food intake are presented in Appendix A.

Adolescents were also classified based on the number of times they reported being satisfied with their BI throughout the study: at all three assessments (*always satisfied*), at two points (*satisfied 2/3*), only on one occasion (*satisfied 1/3*), or never (*never satisfied*). Although the improvement in the DQI-A score over time tended to be greater in boys *always satisfied* with their BI, no significant differences according to BIS trajectories were found (Appendix A).

### 3.3. Body Image Satisfaction and Emotional Management

Mean self-esteem scores decreased significantly over time (Figure 3). Throughout the study, self-esteem was consistently higher in boys and girls satisfied with their BI than in participants dissatisfied with their BI (Table 2). This was reflected in boys and girls dissatisfied with their BI being less likely than BI-satisfied participants to be in the highest self-esteem group (Table 3). The change in mean self-esteem score differed significantly between boys satisfied with their BI and their BI-dissatisfied peers (Figure 3).

Over time, the emotional eating score increased significantly in boys who desired to lose weight and in girls regardless of BIS group (Figure 3). Emotional eating in boys did not differ according to BIS group. In contrast, girls dissatisfied with their BI had significantly higher emotional eating scores than BI-satisfied girls, with girls desiring to gain weight being more likely to be in the highest emotional eating tertile than girls satisfied with their BI (Table 2 and Table 3). Cross-sectional analysis and unadjusted models regarding associations between BIS and emotional management are presented in Appendix A.

The results showed a significant association between the self-esteem score and BIS trajectories, observing that adolescents with lower BIS present a larger decrease in self-esteem (Appendix A). A trend was also observed in emotional eating score, with girls who were never body-satisfied showing the greatest increase in the emotional eating score.

## 4. Discussion

The present study analyzed associations between BIS and nutritional status, DH, and emotional management over a 4-year period in a large sample of adolescents. The analysis shows that a large percentage of adolescents were dissatisfied with their BI, including a high proportion of those without excess weight. Adolescents who desired to gain weight, especially boys, tended to have a less healthy diet. After 4 years of follow-up, the DQI-A improved significantly in body-satisfied adolescents. Although emotional management worsened over time in all BIS groups, it was notably worse (lower self-esteem and higher degree of emotional eating) among body-dissatisfied adolescents, especially girls.

Our results show a higher prevalence of BID than reported in previous studies conducted in several countries with different cultural backgrounds [6,9]. Although BID tends to be more prevalent in adolescents with excess weight [6,44,45,46], in our analysis BID affected a considerable proportion of adolescents without excess weight. This phenomenon has also been observed in adults, especially women [6,9,44], with a higher prevalence of BID consistently reported in women than in men at all stages of life, regardless of nutritional status [4,6,9,12,44,45,46,47]. However, our results show a higher level of BID during adolescence among boys, with boys more frequently desiring to be thickset, whereas girls tended to desire thinner bodies. This may reflect the influence of social stereotypes, with girls idealizing thinner bodies and boys a more muscular physique [4,6,13,48,49].

There are numerous reports in the literature regarding the association between BI and dietary patterns, although most of them focus on adults and/or individuals with eating disorders [6,50,51,52,53]. In line with our findings, several studies have described a relationship between body satisfaction and a healthy diet (eating more fruit and vegetables or having lower odds of following less healthy dietary patterns) [48,54,55,56,57]. Associations have been reported between BID and eating fruit and vegetables and adherence to the Mediterranean diet [58,59,60]. Previous studies also concur with our findings that adolescents desiring to lose weight tend to have a lower intake of unhealthy foods such as sweet beverages, sweets, fast food, processed meat, and snacks [29,56,61,62]. In contrast, those desiring to gain weight tend to have an unhealthy diet characterized by frequent consumption of snacks, fast food, and sweets [29,62,63]. Importantly, these trends identified in cross-sectional analyses are confirmed by the longitudinal results of the present study: the biggest improvement in diet quality was seen in BI-satisfied adolescents, whereas diet quality worsened among those desiring to gain weight. In our study, we also observed that boys showed a greater improvement in the DQI-A compared to girls. However, it is important to note that boys initially showed a poorer food consumption profile. This confirmation that the desire to gain weight is associated with a less healthy diet suggests that food choice in adolescence might be influenced to a greater extent by a desire to conform to normative body shapes rather than a desire to maintain good health.

BID is linked to psychological problems such as low self-esteem and eating disorders independently of nutritional status [44,45]. The decrease in self-esteem during adolescence is influenced by body changes related to puberty, gender identification, self-determination, and self-consciousness [64]. An association between overweight/obesity and low self-esteem is evident in elementary school children and becomes more evident during adolescence [46,65]. Previous studies linked BI in adulthood to distressing experiences related to weight status during childhood and adolescence, such as teasing, marginalization, stigmatization, and discrimination [45,46]. In our study, adolescents satisfied with their BI maintained higher levels of self-esteem than those who were dissatisfied with their BI, and girls tended to score lower than boys, as also reported previously [64]. Given the crucial role of the school environment in the development of self-esteem, it is an ideal setting for the adoption of tailored approaches for boys and girls targeting self-esteem domains such as self-worth, athletic competence, personal mastery, and psychological permeability [64].

The detected association between BID and emotional eating is in line with previous studies. Emotional eating and unhealthy eating patterns associated with BID can increase the risk of binge eating and dieting [12,13,66,67,68], and further longitudinal studies will be needed to characterize gender differences in the association between BID and emotional eating [12,68].

Educational interventions are one of the most widely used tools to prevent unhealthy lifestyle behaviors in children and adolescents. Educational interventions represent one of the most widely used approaches to prevent unhealthy lifestyle behaviors among children and adolescents. However, our findings reinforce that health promotion should prioritize strengthening the link between lifestyle behaviors and overall well-being rather than focusing on weight loss alone [47,66,67]. To achieve this, educational interventions should introduce developmentally appropriate programs beginning before BI attitudes and associated eating and activity patterns become firmly established. The strategy of school-based interventions should encourage a balanced approach to nutrition and exercise that emphasizes overall health and emotional well-being (particularly self-esteem) rather than appearance.

To achieve this, schools should include practical activities, such as guided discussions about media literacy and critical thinking, experiential lessons on recognizing hunger and satiety cues, preparing a weekly nutritionally balanced meal plan, learning what to look for on food labels beyond just calories, designing daily exercise routines to support fitness and well-being, and identifying opportunities for incidental physical activity. Peer-led components, such as cooperative team activities, may further enhance engagement and support self-esteem, given the strong influence of peers on adolescent behavior [69]. The key role of the school environment in health promotion is well known [47,66]; however, further research is needed to determine the best strategy for promoting the acceptance of the BI among adolescents and its relationship to health.

Our results also highlight the relevance of gender-sensitive approaches, as gender stereotypes substantially shape eating behaviors and physical activity patterns. Educational programs should explicitly address how gender stereotypes influence young people’s perceptions of the ‘ideal’ body and associated behaviors. For example, interventions can encourage diversified nutritional choices and challenge the cultural message that muscularity must be pursued through high-protein diets and intensive training [70]. Moreover, educational programs should help adolescents to critically evaluate societal expectations and develop more inclusive perceptions of body diversity. Finally, school-based interventions should counteract the pressure of body shape by promoting self-acceptance, emphasizing functional fitness rather than appearance, and creating supportive environments for participation in a wide range of physical activities.

At the policy level, our findings point to the need for broader structural actions that reinforce the messages delivered in schools. Policies could include mandating age-appropriate BI and media-literacy curricula and supporting teacher training in body-positive health education [71,72]. Nonetheless, exposure to thin-ideal media images has been linked to greater BID and symptoms associated with eating disorders [71]. Governments and educational authorities should also collaborate with media regulators to promote responsible representation of body diversity and reduce the prevalence of unrealistic body standards in youth-targeted media. Furthermore, public health policies should encourage the creation of school and community environments that facilitate daily physical activity for all adolescents, regardless of gender, by providing safe spaces, diverse activity options, and affordable access to sports programs.

### Strengths and Limitations

There are some limitations to this work. First, the study sample was not representative at the national level, since it included only adolescents enrolled in public schools from the metropolitan areas of Madrid and Barcelona, which may restrict the generalizability of the findings. Moreover, because the observational nature of the study, causation cannot be established. Participation in the study was voluntary, therefore some individuals may have been less likely to take part in the study. The participants excluded from the analyses showed some significant sociodemographic differences and a lower diet quality compared to those included (Appendix A). Although the proportion of missing data is relatively small, these differences could have introduced a selection bias since non-respondents might not be missing at random. Other limitations of the study are related to the potential for respondents to adjust responses to coincide with perceived social desirability when reporting food intake or emotional management. Another limitation is that the Stunkard figures and other tools used to assess BIS do not accurately depict all body shapes or distinguish between fat and muscle [4,44,73]. It is also important to consider that adolescents may tend to select the intermediate options on the Stunkard Figure Rating Scale, avoiding the more extreme figures [74]. Moreover, because only a few adolescents in the study population were underweight, they were reclassified into the without excess weight group for BIS analysis, which should be taken into account when interpreting the results. Nevertheless, the main analyses were carried out according to BIS and not according to nutritional status. Given the lack of established cut-off points for the self-esteem and emotional eating scales we used in the study, we classified the adolescents according to their tertile distribution, and thus the direct comparability with other studies may be affected. Lastly, the criteria used to calculate the DQI-A were adjusted to match the items available in the CEHQ and thus differed in some respects from the original proposal [30,31,32,33,34].

A key strength of the study is its evaluation of cross-sectional and longitudinal associations between BI, diet, and emotional management. Whereas most of the available literature focuses on cross-sectional associations in participants with eating disorders, the present study explored the longitudinal relationship between BI and DH and emotional management (self-esteem and emotional eating) in adolescents with no known health conditions. Weight and height were measured by trained nutritionists. The SI! Program for Secondary Schools trial included a large multi-center sample.

## 5. Conclusions

A substantial proportion of adolescents were dissatisfied with their BI, including participants without excess weight. As adolescents grew from 12 to 16, girls without excess weight increasingly showed a desire to lose weight, whereas boys without excess weight increasingly showed a desire to gain weight. BIS was consistently related to food intake during adolescence, with adolescents desiring to gain weight having a less healthy dietary profile. Self-esteem scores decreased between the ages of 12 and 16 regardless of BIS group, but body-satisfied adolescents consistently had the highest self-esteem.

These results suggest that food choice in adolescence might be influenced more by a desire to conform to normative body shapes than by a desire to maintain good health. It is therefore important to develop educational interventions and specific policies regarding advertising and mass media that avoid gender stereotypes and promote self-esteem and body acceptance. These strategies should ideally include a socioemotional-learning approach strengthening the link between food and health regardless of weight status, enhancing media literacy, and avoiding gender stereotypes. Future studies should focus on the longitudinal assessment of the potential link between BIS and other lifestyle behaviors affecting adolescents’ health, such as physical activity, sleep, and screen time, to better understand the possible triggers of BID and develop preventive strategies.

## Figures and Tables

**Figure 1 nutrients-17-03882-f001:**
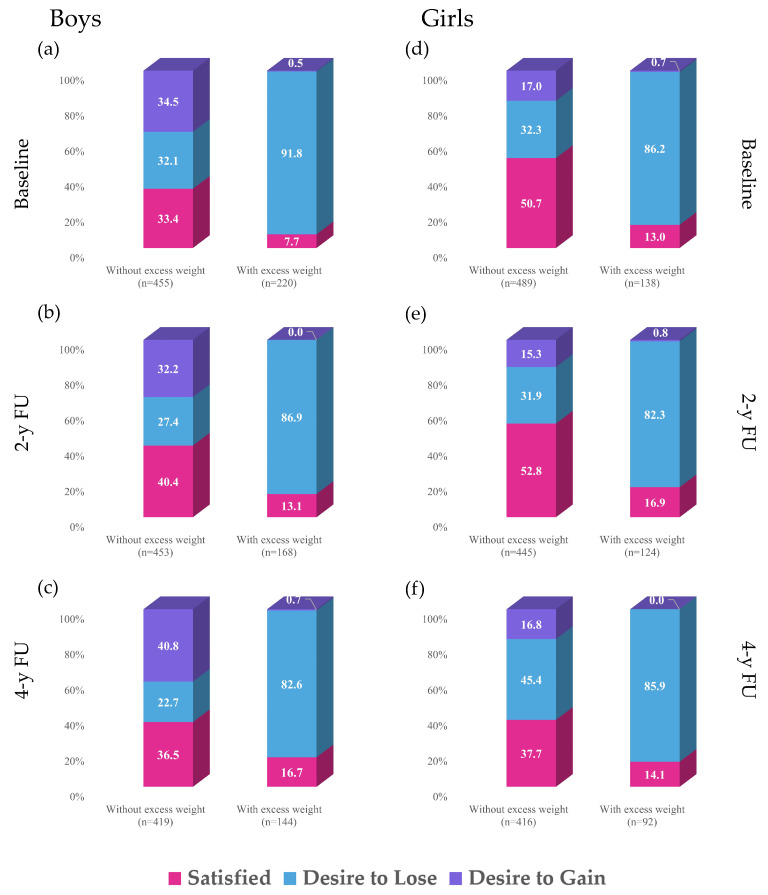
BIS according to nutritional status across adolescence. (**a**) BIS according to nutritional status for boys at baseline; (**b**) BIS according to nutritional status for boys at 2-years FU; (**c**) BIS according to nutritional status for boys at 4-years FU; (**d**) BIS according to nutritional status for girls at baseline; (**e**) BIS according to nutritional status for girls at 2-years FU; (**f**) BIS according to nutritional status for girls at 4-years FU. BIS, body image satisfaction; FU, follow-up, y, years. Nutritional status was defined as age- and sex-adjusted body mass index percentiles according to Centers for Disease Control standards: without excess weight <85th percentile, and with excess weight ≥85th percentile.

**Figure 2 nutrients-17-03882-f002:**
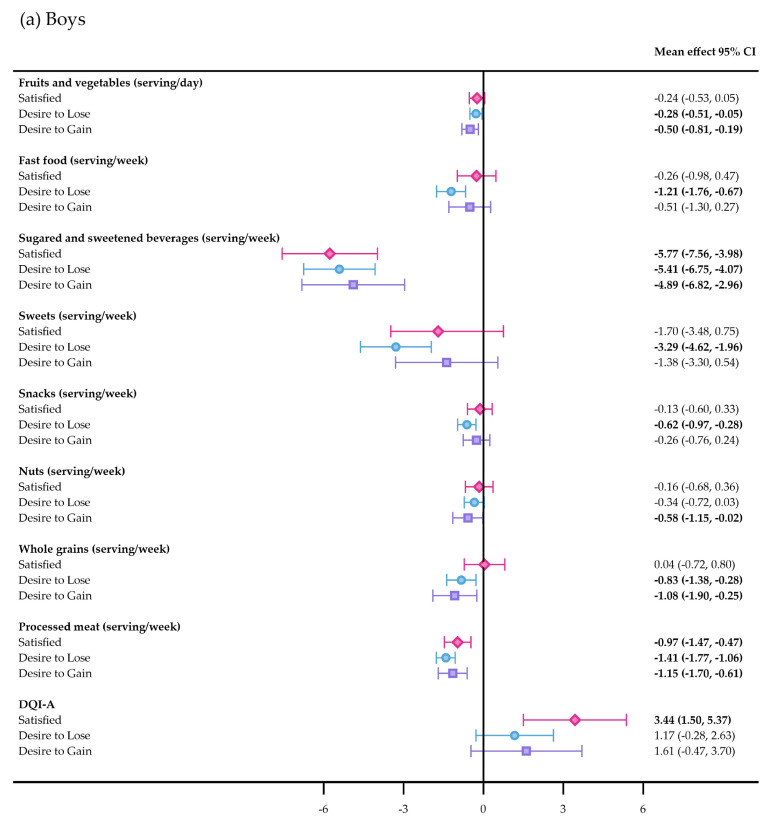
Mean change estimates for food consumption from baseline to 4-year follow-up. DQI-A (ranged from −33/100%), diet quality index for adolescents. The number of participants varied due to data availability. Estimated marginal means (95% confidence intervals (CI)) obtained from multilevel linear mixed-effects models. Fixed effects were body image satisfaction, age, nutritional status (underweight/normal weight/overweight/obesity), educational level (low/intermediate/high/unknown), migrant background (yes/no/unknown), randomization group (long term intervention/short term intervention/control), moderate-to-vigorous physical activity, sexual maturity status, and food consumption (servings/week) at baseline. Region (Madrid/Barcelona) and schools were handled as random effects. Significant differences (*p* ≤ 0.05) are presented in bold. a, significant differences between satisfied vs. desire to lose weight; b, significant differences between desire to lose weight vs. desire to gain weight; c, significant differences between satisfied vs. desire to gain weight.

**Figure 3 nutrients-17-03882-f003:**
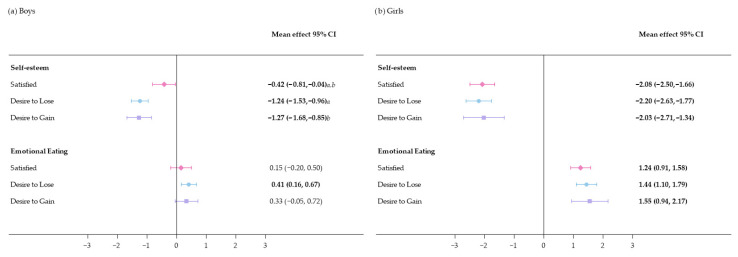
Mean change estimates from baseline to 4-year follow-up for self-esteem and emotional eating. Emotional eating scale ranged 1–12 points, and self-esteem score ranged 1–20 points. The number of participants varied due to data availability. Estimated marginal means (95% confidence intervals (CI)) were obtained from multilevel linear mixed-effects models. Fixed effects were body image satisfaction, age, nutritional status (underweight/normal weight/overweight/obesity), educational level (low/intermediate/high/unknown), migrant background (yes/no/unknown), randomization group (long term intervention/short term intervention/control), moderate-to-vigorous physical activity, sexual maturity status, and self-esteem or emotional eating score at baseline (continuous variable). Region (Madrid/Barcelona) and schools were handled as random effects. Significant differences (*p* ≤ 0.05) are presented in bold. a, significant differences between satisfied vs. desire to lose weight; b, satisfied vs. desire to gain weight.

**Table 1 nutrients-17-03882-t001:** Participant characteristics at baseline by gender.

	Overall	Boys	Girls	*p*-Value
**Gender**, *n* (%)	1315	681 (51.8%)	634 (48.2%)	
**Age, years**, mean (SD)	12.5 (0.4)	12.6 (0.5)	12.5 (0.4)	**0.026**
**Region**, *n* (%)				
Madrid	420 (31.9%)	223 (32.8%)	197 (31.1%)	0.515
Barcelona	895 (68.1%)	458 (67.3%)	437 (68.9%)	
**Migrant background**, *n* (%)				
Spanish	868 (66.0%)	451 (66.2%)	417 (65.8%)	0.696
Migrant background	428 (32.6%)	222 (32.6%)	206 (32.5%)	
Unknown	19 (1.4%)	8 (1.2%)	11 (1.7%)	
**Parental education level**				
Low	245 (18.6%)	116 (17.0%)	129 (20.4%)	0.342
Intermediate	533 (40.5%)	275 (40.4%)	258 (40.7%)	
High	520 (39.5%)	282 (41.4%)	238 (37.5%)	
Unknown	17 (1.3%)	8 (1.2%)	9 (1.4%)	
**Nutritional status**, *n* (%)				
Underweight	38 (2.9%)	22 (3.2%)	16 (2.6%)	**<0.001**
Normal weight	909 (69.6%)	436 (64.3%)	473 (75.3%)	
Overweight	230 (17.6%)	144 (21.2%)	86 (13.7%)	
Obesity	129 (9.9%)	76 (11.2%)	53 (8.4%)	
**Body image satisfaction**, *n* (%)				
Satisfied	437 (33.4%)	169 (25.0%)	268 (42.3%)	**<0.001**
Desire to lose weight	629 (48.0%)	348 (51.4%)	281 (44.4%)	
Desire to gain weight	244 (18.6%)	160 (23.6%)	84 (13.3%)	
**DQI-A**, mean (SD)	57.7 (12.9)	56.7 (13.0)	58.8 (12.7)	**0.004**
Diet quality	42.2 (24.6)	38.9 (25.1)	45.7 (23.6)	**<0.001**
Dietary diversity	78.9 (16.7)	79.7 (16.9)	78.0 (16.4)	0.062
Dietary equilibrium	52.0 (11.8)	51.5 (11.7)	52.6 (11.9)	0.098
**Self-esteem**, mean (SD)	15.6 (1.8)	15.6 (1.8)	15.5 (1.8)	0.221
**Emotional eating**, mean (SD)	4.6 (2.1)	4.6 (2.1)	4.6 (2.0)	0.852

Values are expressed as mean (standard deviation) for continuous variables or as frequency (percentage) for categorical variables. Nutritional status was defined as age- and sex-adjusted body mass index percentiles according to Centers for Disease Control (CDC) standards: underweight <5th percentile; normal weight ≥5th to <85th percentiles; overweight ≥85th to ≤95th percentiles; and obesity >95th percentile. DQI-A, diet quality index for adolescents (ranged from −33/100%). Dietary quality domain ranged from −100/100%, dietary diversity and dietary equilibrium domains ranged from 0/100%, emotional eating scale ranged 1–12 points, and self-esteem score ranged 1–20 points. *p*-values for gender differences were calculated by unpaired t-test or chi-square test, as appropriate. Significant differences (*p* ≤ 0.05) are presented in bold.

**Table 2 nutrients-17-03882-t002:** Food consumption and emotional management by BIS and gender.

REPEATED MEASURES AT BASELINE, 2- AND 4-YEAR FOLLOW-UP						
	**Boys**			**Girls**		
	**Mean (95% CI)**			**Mean (95% CI)**		
	Satisfied	Desire to Lose	Desire to Gain	Satisfied	Desire to Lose	Desire to Gain
**Dietary Habits**						
Fruits and vegetables (serving/day)	2.1 (2.0; 2.3)	2.1 (2.0; 2.3)	2.1 (1.9; 2.3)	2.3 (2.1; 2.5)	2.3 (2.1; 2.5)	2.1 (1.8; 2.4)
Fast food (serving/week)	**4.5 (4.0; 5.1) ^a^**	**4.1 (3.6; 4.7) ^b^**	**5.6 (5.0; 6.2) ^a,b^**	3.8 (3.4; 4.2)	3.6 (3.2; 4.0)	4.1 (3.5; 4.7)
Sugared and sweetened beverages (serving/week)	14.4 (12.9; 16.0)	12.6 (11.1; 14.0)	14.7 (13.0; 16.4)	11.3 (9.3; 13.3)	10.8 (8.8; 12.9)	11.4 (9.1; 13.8)
Sweets (serving/week)	10.1 (9.0; 11.3)	**8.8 (7.7; 9.9) ^b^**	**11.8 (10.5; 13.1) ^b^**	10.4 (8.6; 12.1)	9.7 (8.0; 11.4)	10.9 (8.7; 13.1)
Snacks (serving/week)	2.0 (1.6; 2.5)	**1.9 (1.4; 2.3) ^b^**	**2.4 (2.0; 2.9) ^b^**	2.0 (1.5; 2.6)	1.9 (1.4; 2.5)	2.5 (1.9; 3.1)
Nuts (serving/week)	2.1 (1.7; 2.5)	2.0 (1.6; 2.3)	2.5 (2.1; 2.9)	2.2 (1.5; 2.8)	2.2 (1.5; 2.8)	2.1 (1.4; 2.9)
Whole grains (serving/week)	3.4 (2.4; 4.4)	3.2 (2.2; 4.2)	3.8 (2.8; 4.9)	3.3 (2.6; 3.9)	3.7 (3.1; 4.4)	2.9 (2.0; 3.8)
Processed meat (serving/week)	**4.0 (3.7; 4.4) ^a^**	**3.9 (3.6; 4.2) ^b^**	**4.8 (4.3; 5.2) ^a,b^**	3.9 (3.5; 4.2)	4.2 (3.9; 4.6)	4.6 (4.0; 5.2)
DQI-A	57.2 (55.7; 58.7)	58.3 (56.8; 59.7)	56.8 (55.1; 58.4)	59.4 (58.0; 60.9)	59.8 (58.4; 61.3)	58.4 (56.4; 60.4)
**Emotional Management**						
Self-esteem	**15.5 (15.3; 15.7) ^a,c^**	**14.9 (14.7; 15.2) ^c^**	**15.0 (14.8; 15.2) ^a^**	**14.9 (14.7; 15.2) ^a,c^**	**14.1 (13.9; 14.3) ^c^**	**14.4 (14.0; 14.7) ^a^**
Emotional Eating	4.5 (4.2; 4.8)	4.7 (4.3; 5.0)	4.7 (4.4; 5.0)	**5.0 (4.8; 5.2) ^a,c^**	**5.4 (5.2; 5.6) ^c^**	**5.7 (5.4; 6.0) ^a^**

BIS, body image satisfaction; DQI-A, diet quality index for adolescents (ranged from −33/100%). Emotional eating scale ranged 1–12 points, and self-esteem score ranged 1–20 points. The number of participants varied due to data availability. Estimated marginal means (95% confidence intervals (CI) were obtained from mixed-effect models for repeated measures at baseline, 2- and 4-year follow-up. Fixed effects were body image satisfaction, age, nutritional status (underweight/normal weight/overweight/obesity), educational level (low/intermediate/high/unknown), migrant background (yes/no/unknown), moderate-to-vigorous physical activity, sexual maturity status, and randomization group (long term intervention/short term intervention/control). Region (Madrid/Barcelona), schools, and participants were handled as random effects. Significant differences after Bonferroni correction (*p* ≤ 0.05) are presented in bold: a, significant differences between satisfied vs. desire to gain weight; b, significant differences between desire to lose weight vs. desire to gain weight; c, significant differences between satisfied vs. desire to lose weight.

**Table 3 nutrients-17-03882-t003:** Odds of being in the highest food intake or emotional management category by BIS and gender.

REPEATED MEASURES AT BASELINE, 2- AND 4-YEAR FOLLOW-UP						
	**Boys**			**Girls**		
	**OR (95% CI)**			**OR (95% CI)**		
	Satisfied	Desire to Lose	Desire to Gain	Satisfied	Desire to Lose	Desire to Gain
**Dietary Habits**						
Fruits and vegetables	[Ref]	1.10 (0.76; 1.61)	0.86 (0.58; 1.29)	[Ref]	0.89 (0.63; 1.27)	0.75 (0.45; 1.23)
Fast food	-	0.85 (0.58; 1.24)	**1.77 (1.21; 2.59)**	-	0.79 (0.56; 1.11)	1.39 (0.87; 2.20)
Sugared and sweetened beverages	-	**0.54 (0.37; 0.79)**	0.98 (0.67; 1.42)	-	0.75 (0.53; 1.04)	1.09 (0.69; 1.72)
Sweets	-	**0.64 (0.44; 0.92)**	**1.86 (1.29; 2.68)**	-	0.73 (0.52; 1.00)	1.21 (0.79; 1.88)
Snacks	-	0.97 (0.59; 1.62)	**1.85 (1.13; 3.03)**	-	0.75 (0.48; 1.18)	1.42 (0.81; 2.49)
Nuts	-	0.92 (0.60; 1.39)	1.15 (0.75; 1.77)	-	0.82 (0.57; 1.18)	0.91 (0.55; 1.51)
Whole grains	-	1.03 (0.72; 1.48)	0.99 (0.67; 1.45)	-	1.20 (0.87; 1.65)	0.77 (0.48; 1.24)
Processed meat	-	0.85 (0.59; 1.24)	1.37 (0.94; 2.00)	-	1.23 (0.87; 1.73)	1.19 (0.74; 1.92)
DQI-A	-	1.25 (0.83; 1.89)	0.85 (0.55; 1.31)	-	1.16 (0.79; 1.71)	0.92 (0.54; 1.58)
**Emotional Management**						
Self-esteem	[Ref]	**0.57 (0.40; 0.81)**	**0.66 (0.46; 0.95)**	[Ref]	**0.44 (0.31; 0.63)**	**0.45 (0.26; 0.77)**
Emotional Eating	-	1.27 (0.85; 1.88)	1.45 (0.96; 2.19)	-	1.20 (0.85; 1.70)	**2.06 (1.28; 3.32)**

BIS, body image satisfaction; DQI-A, diet quality index for adolescents. The number of participants varied due to data availability. Odds ratios (OR) (95% CI) of being in the highest consumption tertile or for being in the highest emotional eating or self-esteem categories were calculated by logistic regression. Fixed effects were body image satisfaction, age, nutritional status (underweight/normal weight/overweight/obesity), educational level (low/intermediate/high/unknown), migrant background (yes/no/unknown), moderate-to-vigorous physical activity, sexual maturity status, and randomization group (long term intervention/short term intervention/control). Region (Madrid/Barcelona), schools, and participants were handled as random effects. Significant differences (*p* < 0.05) are presented in bold. High food consumption, emotional eaters and participants with high self-esteem were defined as those in the highest gender-specific tertile of food consumption, DQI-A (ranged from −33/100%), emotional eating (ranged 1–12 points) and self-esteem score (ranged 1–20 points), respectively.

## Data Availability

Data availability to external researchers is restricted to related project proposals upon request to the corresponding author. Based on these premises, deidentified participant data will be available with publication after approval of the proposal by the Steering committee and a signed data sharing agreement.

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
