# Peer review of "Body Image Satisfaction, Food Consumption, Diet Quality, and Emotional Management in Adolescence: A Longitudinal Analysis from the SI! Program for Secondary Schools Trial"

_nutrients, 2025, doi:10.3390/nu17243882_

Round 1

Reviewer 1 Report

Comments and Suggestions for Authors

Overall, I consider the manuscript to be very well designed and clearly written, and I believe it makes a valuable contribution to the field. 

However, I would like to point out one aspect that I consider important in the analysis. I could not find information on key sociodemographic variables such as socioeconomic status, type of school (public, state-subsidised, or private), or area of residence. This information does not appear to be included in Supplementary Table 9 either.

Given the well-established influence of these variables on dietary patterns, lifestyle behaviours, and health outcomes, I suggest that the authors clarify whether these data were collected and, if so, why they were not included in the analyses. If these variables were not available, I believe this should be explicitly acknowledged as a limitation of the study.

Reviewer 2 Report

Comments and Suggestions for Authors

This article describes the results of a four-year cohort study conducted as part of the SI! program in Spain, examining the relationship between body image satisfaction (BIS), eating habits, diet quality (DQI-A), and emotional aspects (self-esteem and emotional eating) in adolescents aged 12–16. The study included 1,315 adolescents, assessed at three time points. The results suggest that BIS is associated with diet and emotions, and the direction of these associations differs by gender. The manuscript is well prepared, however needs some major revisions: 
1. Lack of full control for potential confounding variables
Despite advanced modeling (mixed models), physical activity level, pubertal status, and mental health status (e.g., anxiety, depression), which can strongly influence BIS, emotions, and eating habits, were not considered.
2. Insufficient discussion of the impact of participation in the SI! program. Participants were from the intervention and control groups of the SI! program, which could potentially have influenced the BIS, DQI-A, and self-esteem scores.
3. Potential overinterpretation of causality
The findings suggest causal relationships (e.g., "desire to gain weight was associated with an unhealthy diet"), despite the observational nature of the analysis.
4. Insufficient analysis of change over time
Although models for repeated data were used, the full trajectory of change in BIS, DQI-A, self-esteem, and emotional eating by group was not shown.
5. Small group of underweight and obese individuals
The group with a BMI <5th percentile and >95th percentile was relatively small, yet it provided a reference point for interpretation.
6. Lack of references to precise cutoffs for "emotional eating" and "low self-esteem"
The use of tertiles instead of established cutoffs complicates comparison with other studies.

Reviewer 3 Report

Comments and Suggestions for Authors

Dear Authors,

I would like to congratulate you on conducting a valuable study with a robust longitudinal design and rigorous methodology. I recommend acceptance of your manuscript following minor revisions addressing the points outlined below.

Please revise the following aspects:

  1. The exclusion of 11 participants (those with eating disorders, emotional disorders, and one undergoing gender transition) lacks sufficient justification in the current manuscript.
  • Methods section: Please provide a clear rationale explaining why the inclusion of these participants would compromise the validity of your findings.

  1. While your conclusions are sound, the practical recommendations remain somewhat general and would benefit from greater specificity.

Required revisions:

Please expand the practical implications section to include more concrete and actionable recommendations:

-For educational interventions

-Gender-differentiated approaches

-At the policy level

Reviewer 4 Report

Comments and Suggestions for Authors

This study aimed to describe body image satisfaction in adolescents in relation to their nutritional status, to assess the associations between body image satisfaction and food intake, and to explore the interplay between body image satisfaction and emotional management. In this prospective cohort study, adolescents were followed up from 2017 to 2021. A total of 1326 adolescents were assessed at the beginning of 1st grade of Secondary School (baseline, ~12 years old). Follow-up assessments after 2 years (~14 years old) and 4 years (~16 years old) were performed at the end of 2nd and 4th grade, respectively. It was established that body image satisfaction was associated with nutritional status and emotional management, showing gender differences. The desire to gain weight was associated with unhealthier dietary habits. It was concluded that educational interventions should promote self-esteem and body image satisfaction by focusing messages on healthy eating instead of body weight.

The Introduction section is rather short and does not emphasize enough the importance of the issues that have been studied within this research. The authors should improve this part of the manuscript following the previously mentioned suggestion.

This topic is surely important, but it cannot be said that it is original, since there are recent studies dealing with the studied issues elsewhere in the world. Body image satisfaction has indeed been little studied in relation to its longitudinal effects on dietary habits and emotional management, and the present study addresses this gap. Because of the latter, it can be said that this research is relevant to the field and would probably be of interest to the readers of the Nutrients journal.

A key contribution of this study is its focus on the longitudinal relationship between body image and dietary habits and emotional management (self-esteem and emotional eating) in adolescents with no known health conditions. This is highly valuable since most of the available literature has predominantly focused solely on cross-sectional associations in participants with eating disorders.

Although the methodology is very well described, the authors should clearly state that this was a prospective cohort study and not only cite the previously published articles.

If authors are saying “There are numerous reports in the literature regarding the association between BI and dietary patterns,” it is not appropriate that they have cited only one article “[6]" and this article is a systematic review done by the authors of this manuscript!! This should be corrected.

The Conclusions are consistent with the evidence and arguments presented, and they address the main question posed.

Round 2

Reviewer 1 Report

Comments and Suggestions for Authors

I appreciate the opportunity to review the revised version of your manuscript, and I believe that the changes introduced have clearly improved the quality of the study.

It is now also clearly specified that the participating schools are public schools. In the Spanish context, the geographical and socio-environmental background of students is an important factor, as it is closely linked to cultural, social, and economic differences that may influence the study outcomes.

Overall, I consider that the manuscript is methodologically sound and suitable for publication.

Reviewer 2 Report

Comments and Suggestions for Authors

The Authors have revised the manuscript.